# The Impacts of Data, Ordering, and Intrinsic Dimensionality on Recall in Hierarchical Navigable Small Worlds

## ABSTRACT

Vector search systems, pivotal in AI applications, often rely on the Hierarchical Navigable Small Worlds (HNSW) algorithm. However, the behaviour of HNSW under real-world scenarios using vectors generated with deep learning models remains under-explored. Existing Approximate Nearest Neighbours (ANN) benchmarks and research typically has an over-reliance on simplistic datasets like MNIST or SIFT1M and fail to reflect the complexity of current use-cases. Our investigation focuses on HNSW's efficacy across a spectrum of datasets, including synthetic vectors tailored to mimic specific intrinsic dimensionalities, widely-used retrieval benchmarks with popular embedding models, and proprietary e-commerce image data with CLIP models. We survey the most popular HNSW vector databases and collate their default parameters to provide a realistic fixed parameterisation for the duration of the paper.

We discover that the recall of approximate HNSW search, in comparison to exact K Nearest Neighbours (KNN) search, is linked to the vector space's intrinsic dimensionality and significantly influenced by the data insertion sequence. Our methodology highlights how insertion order, informed by measurable properties such as the pointwise Local Intrinsic Dimensionality (LID) or known categories, can shift recall by up to 12 percentage points. We also observe that running popular benchmark datasets with HNSW instead of KNN can shift rankings by up to three positions for some models. This work underscores the need for more nuanced benchmarks and design considerations in developing robust vector search systems using approximate vector search algorithms. This study presents a number of scenarios with varying real world applicability which aim to better increase understanding and future development of ANN algorithms and embedding models alike.

## CCS CONCEPTS

• **Information systems** → **Retrieval effectiveness**; *Relevance assessment*; *Search engine indexing*.

## KEYWORDS

HNSW, ANN, Information Retrieval

**ACM Reference Format:**
Anonymous Author(s). 2024. The Impacts of Data, Ordering, and Intrinsic Dimensionality on Recall in Hierarchical Navigable Small Worlds. In *Proceedings of Make sure to enter the correct conference title from your rights confirmation emai (ICTIR 2024)*. ACM, New York, NY, USA, 9 pages. https://doi.org/XXXXXXX.XXXXXXX

## 1 INTRODUCTION

The efficient retrieval of nearest neighbours in high-dimensional spaces is a requirement for many Artificial Intelligence (AI) applications. This need has driven the development and widespread adoption of Approximate Nearest Neighbours (ANN) algorithms, among which the Hierarchical Navigable Small Worlds (HNSW)[28] algorithm has emerged as a preeminent choice for search and recommendation applications.

Despite its extensive utilization, the behaviour and performance of the HNSW algorithm under real-world conditions remains insufficiently explored. This gap in understanding is particularly critical given the evolving nature of datasets in contemporary AI applications. Existing benchmarks for evaluating ANN systems, which often rely upon simplistic or lower-dimensional datasets (MNIST[13], SIFT1M[21], etc.), do not adequately reflect many popular real-world use-cases. These datasets are highly curated and do not contain vectors from machine learning embedding models. This discrepancy raises questions about the applicability and reliability of these benchmarks in guiding the design and implementation of vector search systems in real-world scenarios.

To bridge the gap between benchmarks and contemporary applications, our research studies the behaviour of HNSW search across vector spaces produced with various methods including synthetic data, popular retrieval benchmarks with popular text embedding models, and real-world e-commerce data with multimodal embeddings from CLIP[35] models.

We collate a survery of popular HNSW vector search systems and their default parameters to provide a fixed parameterisation of the algorithm, unlike prior research we study the behaviour of HNSW as a function of data, models, and indexing conditions rather than of parameterisation. This methodology allows us to identify a relationship between both the intrinsic dimensionality of vector spaces as a whole as well as the Local Intrinsic Dimensionality (LID) of vectors within the dataset and the order in which they are added. By controling insertion order of data with LID we observe recall dropping by as much as 12.8 percentage points. We extend this observation about ordering to real world applications and quantify the difference in intrinsic dimensionality between a dataset and its constituent categories. By indexing data with different orders of categories we are able to vary recall by up to 8 percentage points.

Through this work, we aim not only to contribute valuable insights into the HNSW algorithm's behaviour, but also to challenge the prevailing benchmarks for ANN search, encouraging the development of more robust and reliable vector search systems.

## 1.1 Contributions

In this work we provide a survey of existing vector search systems that use HNSW and the parameters that they provide as default. Many of these default parameterisations are not documented or easily accessible and are only mentioned in code. We provide the default parameters at time of writing in this work (Table 1).

In addition to the findings in this work we also release a large 500GB collection of all the vectors used in this research[1]. This includes the complete embeddings from seventeen popular open-source models on the 7 datasets used in this work as well as MS-MARCO [5]. To compliment these embeddings we provide the pointwise Local Intrinsic Dimensionality (LID) estimates for every vector with its 100 nearest neighbours (which is a time consuming $O(n^2)$ task).

## 2 RELATED WORK

The impacts of intrinsic dimensionality, specifically LID, have been studied by Aumüller et al[4]. In their work it was identified that query sets of varying difficulty would be constructed by identifying the LID of the queries, the impact of this being that averaging results across all queries could mask behaviour of the algorithms in benchmarking.

In the source code for the implementation of the original paper, HNSWLib. The authors make reference to a relationship between the $M$ parameter and the intrinsic dimensionality of the data, stating that a higher $M$ of 48-64 is required for good recall on data with higher intrinsic dimensionality. However, this relationship is not considered in detail in the original work and does not appear to be quantified in research[28].

In other work, P. Lin et al analyse the search time behaviour of HNSW on data with varying LIDs and identify that the hierarchical component of HNSW offers less benefit over a flat search as the LID of the data increases. If the graph contains close neighbourhoods with minimal intersection between their nearest neighbour lists, the search can find it difficult to jump from one local minima to another. This results in worse recall for given parameters, or worse latency for a given recall, as the LID of the data increases [27].

The "curse of dimensionality" is widely acknowledged as a feature which impacts the efficacy of retrieval systems and measures of similarity in general. However, data with a large apparent dimensionality can often have a low intrinsic dimensionality. For some KNN algorithms such as KD-Trees, a sufficiently small intrinsic dimensionality can reduce the likelihood of a low quality solution. For other KNN algorithms a lower intrinsic dimensionality can be indicative of potentially favourable performance with dimensionality reduction techniques applied. [8].

## 3 THE HNSW ALGORITHM AND ITS IMPLEMENTATIONS

For this research we focus on more pure implementations of the HNSW algorithm to avoid introducing additional variables into the experiments. The HNSW implementations from FAISS[14, 22] and HNSWLib[28] are used in this paper as they create a single HNSW graph for all data and do not have any complexities around sharding and segmentation for horizontal scaling and mutability[2].

## 3.1 HNSW Parameters

The HNSW algorithm has three primary parameters that impact recall, memory, and latency: $M$, $efConstruction$, and $efSearch$. $M$ is the number of bidirectional links to form between each node in the graph, the final layer of the graph typically uses $2 \cdot M$ links; this impacts the recall, memory usage, and latency where higher $M$ gives better quality retrieval but worse performance. $efConstruction$ is the number of candidates to hold in the heap when constructing the graph, evaluating more candidates gives better graphs with higher recall, however it does increase the time spent indexing; $efConstruction$ does not impact search latency. $efSearch$ is the size of the candidate list to hold in the heap at search time, higher $efSearch$ can increase recall at the cost of latency.

*Methodology for Determining Parameters and Reasoning.* We fix the HNSW graph parameters for all experimentation. We acknowledge that many challenges regarding recall for approximate nearest neighbours with HNSW can be circumnavigated by increasing $M$, $efConstruction$, and/or $efSearch$. However, in reality it is not feasible to extensively search the parameter space for optimal parameters, and furthermore, it is not feasible to scale these parameters beyond a point as latency degrades.

To determine appropriate fixed defaults for this experimentation, we surveyed approximate nearest neighbours systems that use HNSW to determine their defaults.

**Table 1: Default Settings of Various Vector Databases (Approximate Nearest Neighbours Systems)**

| System | $M$ | $efConstruction$ | $efSearch$ |
|---|---|---|---|
| MarqoV1[32] | 16 | 128 | $k$ |
| HNSWLib[16] | 16 | 200 | 10 |
| FAISS[38] | 32 | 40 | 16 |
| Chroma[10] | 16 | 100 | 10 |
| Weaviate[49] | 64 | 128 | 100 |
| Qdrant[34] | 16 | 100 | 128 |
| Milvus[29] | 18 | 240 | No Default |
| Vespa[42] | 16 | 200 | $k$ |
| Opensearch (nmslib)[32] | 16 | 512 | 512 |
| Opensearch (Lucene)[32] | 16 | 512 | $k$ |
| Elasticsearch (Lucene)[15] | 16 | 100 | No Default |
| Redis[37] | 16 | 200 | 10 |
| PGVector[33] | 16 | 64 | 40 |

Note: In this table, $k$ represents the number of results to return. Systems with $efSearch = k$ do not specify a default $efSearch$ and set it to $k$ at seach time.

It is clear that many HNSW implementations rely upon relatively low defaults for the algorithm parameters[3]. It is clear that $M = 16$ is

---

[1]The data can be accessed via Hugging Face here: https://huggingface.co/datasets/anonymous/benchmark-embeddings

[2]Modern example include Lucene and Vespa which include production features like sharding and replicas.

[3]Parameters displayed here are current at time of writing, the parameters displayed at some cited URLs are subject to change with time.

widely accepted as a sensible default parameter for the number of bidirectional links to form in the graph, 16 is the median. Values for $efConstruction$ vary more widely ranging from 40 to 512, for the work we opt to fix it at $efConstruction = 128$ as this is the median. $efSearch$ is more complicated as a number of the implementations either do not provide a default or use the number of results to return ($k$) as the value for $efSearch$, thus it is use case dependant. As such, we set $efSearch$ as the median of the parameters observed in industry when $k$ for the given task is substituted into Table 1.

## 4 DATASETS

In this section we describe the three main groups of datasets used in this study.

### 4.1 Synthetic Data

In the synthetic case, we consider artificially generated vectors to control their properties, particularly their intrinsic dimensionality. To generate vectors of varying intrinsic dimensionality, we increase their complexity by varying the number of orthonormal basis vectors used in their construction. The Gram-Schmidt algorithm is used to create an orthonormal basis, and varying numbers of these orthonormal basis vectors are then combined to form datasets of vectors with varying intrinsic dimensionalities.

*Gram-Schmidt Orthonormalization.* Let $\mathbf{V} = \{\mathbf{v}_1, \mathbf{v}_2, \ldots, \mathbf{v}_k\}$ be a set of $k$ randomly generated vectors in $\mathbb{R}^d$, where $d$ represents the dimensionality of the space. The Gram-Schmidt process is applied to these vectors to obtain an orthonormal basis $\mathbf{U} = \{\mathbf{u}_1, \mathbf{u}_2, \ldots, \mathbf{u}_k\}$, where each $\mathbf{u}_i$ is defined recursively by:

$$\mathbf{u}_i = \frac{\mathbf{w}_i}{\|\mathbf{w}_i\|} \quad \text{with} \quad \mathbf{w}_i = \mathbf{v}_i - \sum_{j=1}^{i-1} \text{proj}_{\mathbf{u}_j}(\mathbf{v}_i)$$

and the projection of $\mathbf{v}_i$ onto $\mathbf{u}_j$ is given by:

$$\text{proj}_{\mathbf{u}_j}(\mathbf{v}_i) = \frac{\langle \mathbf{v}_i, \mathbf{u}_j \rangle}{\langle \mathbf{u}_j, \mathbf{u}_j \rangle} \mathbf{u}_j$$

*Generating Data with Intrinsic Dimensionality of $k$.* Once the orthonormal basis $\mathbf{U}$ is established, synthetic data $\mathbf{X}$ can be generated. This involves creating $n$ linear combinations of the basis vectors, where $n$ is the number of desired data vectors. We first define $\mathbf{C}$, an $n \times k$ matrix whose entries $c_{ij}$ are coefficients drawn from a normal distribution. Each data vector $\mathbf{x}_i$ is constructed as $\mathbf{x}_i = \sum_{j=1}^{k} c_{ij}\mathbf{u}_j$. Thus, the data matrix $\mathbf{X}$ in $\mathbb{R}^{n \times d}$ is represented by $\mathbf{X} = \mathbf{CU}$ where $\mathbf{U}$ is a $k \times d$ matrix containing the orthonormal basis vectors. Each row of $\mathbf{X}$ represents a data vector in the space spanned by the basis $\mathbf{U}$. In practice this creates a dataset of $n$ unique random vectors with dimension $d$ which exist in a vector space of intrinsic dimensionality $k$.

### 4.2 Retrieval Datasets

Text embedding models have been widely benchmarked for retrieval on a number of popular standard benchmark datasets. One popular aggregation of retrieval evaluations for text embedding models is the Massive Text Embedding Benchmark (MTEB)[30]. For this work we select a subset (see Table 2) of the datasets used for evaluation in

MTEB as well as a selection of the most popular and best performing models at the time of this research.

The datasets in Table 2 are used in this work.

**Table 2: Standard retrieval benchmark datasets used.**

| Dataset | No. Queries | Corpus Size | Task Type |
|---|---|---|---|
| NFCorpus[7] | 323 | 3.6K | Asymmetric |
| Quora | 10k | 523k | Symmetric |
| SCIDOCS[11] | 1k | 25k | Asymmetric |
| SciFact[45] | 300 | 25k | Asymmetric |
| CQADupstack[17] | 13.1k | 547k | Asymmetric |
| TRECCOVID[43] | 50 | 171k | Asymmetric |
| ArguAna[44] | 1.4k | 8.7k | Symmetric |

The datasets fall into one of two task types:

- **Asymmetric:** The queries and corpus documents are asymmetric. Queries are questions or shorter statements used for retrieving related documents and answers from the corpus;
- **Symmetric:** The queries and corpus documents are the same type of text. The goal is to find text in the corpus which is similar (for example, in the Quora dataset, the task is to find titles that are similar to the query title)

For each of the datasets, embeddings are created with the models in Table 3.

**Table 3: Models used to embed the standard retrieval benchmark datasets.**

| Model | Embedding Dimension |
|---|---|
| bge-base-en[50] | 768 |
| bge-small-en[50] | 384 |
| bge-base-en-v1.5[50] | 768 |
| bge-small-en-v1.5[50] | 384 |
| stella-base-en-v2[2] | 768 |
| e5-base[46] | 768 |
| e5-small[46] | 384 |
| e5-base-v2[46] | 768 |
| e5-small-v2[46] | 384 |
| multilingual-e5-large[47] | 1024 |
| multilingual-e5-base[47] | 768 |
| multilingual-e5-small[47] | 384 |
| ember-v1[36] | 1024 |
| all-MiniLM-L6-v2[48] | 384 |
| bge-micro[3] | 384 |
| gte-base[26] | 768 |

The vector spaces produced for each model and dataset combination have their own properties regarding intrinsic dimensionality and local intrinsic dimensionality which are studied in this work.

### 4.3 Real World Datasets

In addition to the synthetic data and standard benchmark retrieval datasets we also verify our findings under real-world conditions.

Our real-world datasets are a proprietary collection of product images. We present two datasets:

- An e-commerce catalogue of collectibles, handbags, streetwear, sneakers, and watches; and
- A homewares catalogue of home, furniture, kitchenware, wall, renovation, bed, rugs, lighting, baby, lifestyle, pet, and office.

To assess the applicability of our findings in a real-world setting we leverage the relationship between intrinsic dimensionality and categories in online retail catalogues. Items belonging to one category have their own intrinsic dimensionality which is lower than that of the entire dataset. To evaluate HNSW on this data, we use permutations of the categories to form insertion orders for data into the HNSW indexes; recall is computed for each order.

## 5 EVALUATION METHODOLOGY AND RESULTS

For the purposes of this work we define recall as the number of documents returned by an exact retriever which are also retrieved by an approximate one, in this case, KNN as the exact retriever and HNSW as the approximate retriever. Formally, for an approximate retriever (A) and an exact retriever (E) within a dataset $X$ using a set of queries $Q$ where $k$ results are retrieved for each query, the recall at $k$ is defined as follows:

Let $R_A(q, k)$ denote the set of $k$ results retrieved by the approximate retriever $A$ from $X$ for a query $q \in Q$, and $R_E(q, k)$ denote the set of $k$ results retrieved by the exact retriever $E$ from $X$ for the same query $q$. The recall for a single query $q$, is defined as the fraction of relevant documents retrieved by the approximate retriever $A$ out of the relevant documents retrieved by the exact retriever $E$.

$$\bar{recall}(Q, k) = \frac{1}{|Q|} \sum_{q \in Q} \frac{|R_A(q, k) \cap R_E(q, k)|}{|R_E(q, k)|}$$

where $|Q|$ is the number of queries in the set $Q$. Unless otherwise stated, recall is calculated at $k = 10$.

### 5.1 Evaluating Recall on Synthetic Vectors

As described in section 4.1, the synthetic data consists of vectors of arbitrary intrinsic dimensionality which are created by combining varying numbers of orthonormal basis vectors. To assess the qualities of these vectors the intrinsic dimensionality is quantified with a Principal Component Analysis (PCA) based approach.

*Estimation of Intrinsic Dimensionality Using PCA.* The intrinsic dimensionality of a dataset can be estimated using PCA by identifying the number of principal components that capture a significant proportion of the total variance in the dataset. Let $X$ be a dataset and $\lambda_i$ represent the explained variance ratio of the $i$-th principal component in the PCA. We compute a PCA on the dataset $X$ to obtain the explained variance ratios $\lambda_1, \lambda_2, \ldots, \lambda_n$, where $n$ is the total number of features in $X$. The sum of the explained variance ratios for $k$ components is defined as $C(k) = \sum_{i=1}^{k} \lambda_i$. To determine the intrinsic dimensionality we find the smallest number $k$ such that the cumulative sum $C(k)$ is greater than or equal to a pre-defined threshold $\theta$ (e.g., $\theta = 0.99$ for 99% variance). The value of $k_{\text{intrinsic}}$

represents the estimated intrinsic dimensionality of the dataset $X$ and describes the minimum dimensionality within the data which captures the specified proportion of variance $\theta$ in the data. For our work, $\theta = 0.99$.

*Evaluation of Recall on Synthetic Data.* We can verify the process used to generate this data by visualising the cumulative sum of explained variance ratio from the PCA for datasets constructed with varying numbers of orthonormal basis vectors as shown in Figure 1.

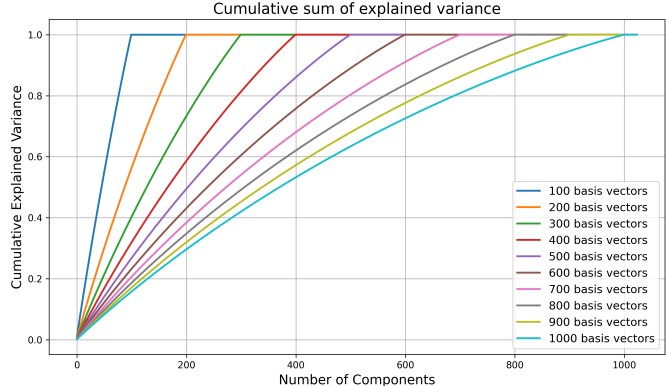

**Figure 1: Cumulative sum of explained variance ratio from a PCA on datasets with 1024 dimensional vectors constructed from varying numbers of orthonormal basis vectors.**

What we observe is that as the number of orthonormal basis vectors used to generate the synthetic data increases, the recall achieved with both HNSWLib and FAISS decreases. Figure 2 depicts the recall for HNSWLib and FAISS as the number of orthonormal basis vectors used to construct the data increases. The number of orthonormal basis vectors used to construct the data is the same for the indexed vectors and the query vectors.

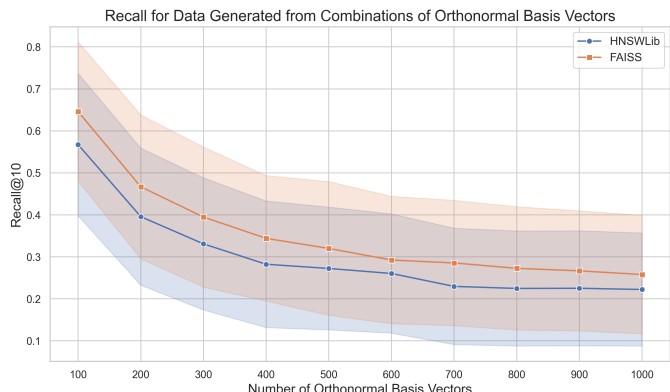

**Figure 2: Recall for HNSWLib and FAISS at $efConstruction = 128$, $M = 16$, and $efSearch = 40$ on a dataset of 10,000 vectors with 1,000 queries.**

## 5.2 Popular Models on Benchmark Datasets

The synthetic data evaluation from section 5.1 shows that there exist properties of the vector space which can directly influence recall of HNSW for a given parameterisation. It follows that models whose vector spaces exhibit different properties for a given dataset can also impact recall. Many popular retrieval models are trained with some form of contrastive loss which provides no explicit control for properties such as the intrinsic dimensionality or local intrinsic dimensionality. Furthermore, training and evaluation of these models is typically only done in the context of exact KNN.

Evaluation on benchmark datasets outlined in Table 2 for all models identified in Table 3 shows that rankings of models change when evaluated with various retrieval systems. This is to say that a retrieval leaderboard established with exact KNN is not perfectly representative of one produced using approximate nearest neighbours retrieval. Models are ranked using Normalised Discounted Cumulative Gain (NDCG)[23].

**Table 4: Average NDCG@10 with $efSearch = 10$ comparing change in performance for different retrievers. Sorted by descending exact NDCG@10.**

| Model | NDCG@10 Exact | NDCG@10 HNSWLib | NDCG@10 FAISS | Rank Change (HNSWLib/FAISS) |
|---|---|---|---|---|
| ember-v1 | 0.4318 | 0.4171 | 0.4043 | 0 / 0 |
| bge-base-en-v1.5 | 0.4275 | 0.4073 | 0.3922 | -1 / -1 |
| gte-base | 0.4244 | 0.4093 | 0.3991 | 1 / 1 |
| bge-base-en | 0.4180 | 0.3920 | 0.3748 | -2 / -2 |
| stella-base-en-v2 | 0.4155 | 0.3974 | 0.3862 | 1 / 1 |
| bge-small-en-v1.5 | 0.4120 | 0.3925 | 0.3791 | 1 / 1 |
| bge-small-en | 0.4011 | 0.3758 | 0.3600 | 0 / -1 |
| e5-base-v2 | 0.3965 | 0.3707 | 0.3495 | -1 / -1 |
| e5-base | 0.3964 | 0.3592 | 0.3479 | -1 / -1 |
| all-MiniLM-L6-v2 | 0.3886 | 0.3712 | 0.3634 | 2 / 3 |
| multilingual-e5-large | 0.3868 | 0.3555 | 0.3405 | 0 / 0 |
| e5-small-v2 | 0.3819 | 0.3404 | 0.3122 | -1 / -3 |
| e5-small | 0.3771 | 0.3403 | 0.3233 | -1 / 0 |
| multilingual-e5-base | 0.3738 | 0.3372 | 0.3182 | -1 / 0 |
| bge-micro | 0.3667 | 0.3411 | 0.3278 | 3 / 3 |
| multilingual-e5-small | 0.3638 | 0.3308 | 0.3122 | 0 / 0 |

In Table 4 we observe that ranks shift up and down by up to three places when evaluated with different retrieval systems, these experiments use $efSearch = 10$ at $k = 10$. Smaller models like all-MiniLM-L6-V2 and bge-micro see improvements in relative performance when used in approximate retrieval systems, moving up the leader board by 2-3 places at $efSearch = 10$.

### 5.2.1 Local Intrinsic Dimensionality and Popular Benchmark Datasets.
The analysis presented in section 5.1 reveals a significant relationship between the intrinsic dimensionality of the dataset and the recall performance of the system, with recall falling by approximately 50% as synthetic data approaches full rank. This observation leads us to hypothesise that HNSW graphs exhibit enhanced performance when they are structured in a manner that increases the probability of selecting entry points at each layer that are proximally located to any region within the graph. By proactively assessing the pointwise

LID of data vectors, we can strategically influence the construction of the graph to optimise (or impair) its recall.

In particular, constructing a graph with data sorted in descending order of LID appears to mimic a process similar to simulated annealing. This approach facilitates the late integration of clusters characterized by low LID values. Conversely, graphs initialized with data in ascending order of LID suffer from early establishment of tight localities in the graph comprised of low LID vectors. This setup deteriorates the initial conditions for graph optimization, leaving the integration of high LID vectors until the end stages. For the smaller datasets (arguana, nfcorpus, scifact, and scidocs) we are able to calculate the average path length of the final layer of the graphs constructed with HNSWLib. A Pearson correlation coefficient of 0.61 is observed between recall and average path length for these datasets across all models, this positive relationship indicates that longer path lengths yield better recall, this also aligns with the hypothesis that inserting high LID data first delays integration of tight clusters within the graph. The computation of pointwise LIDs was conducted using a Maximum Likelihood Estimation (MLE) method considering the 100 exact nearest neighbours[25].

### 5.2.2 LID Ordered Insertion and Recall.
The recall@10 was calculated for every model on the test sets of the 7 standard benchmark datasets identified in Table 2, the recall for each of the models was then averaged across all datasets. Results are presented in Table 5, bold values indicate the highest recall that was achieved for HNSWLib or FAISS. When ordered by LID the recall is affected with consistent patterns, on average HNSWLib and FAISS implementations achieve 2.6 and 6.2 percentage points better recall respectively when data is inserted in descending LID order at $efSearch = 10$.

Table 5 shows that for all but one model (gte-base with FAISS) the recall was higher when inserting data in order of descending local intrinsic dimensionality. A random insertion order is consistently in between the ascending and descending LID insertion orders, reinforcing the hypothesis that the LID can be used to strategically influence the recall of the system, potentially towards an upper and lower extreme. The change in recall for each order varies significantly from the HNSWLib implementation to the FAISS implementation with a maximum difference between the two orders of 5.6 percentage points for HNSWLib (e5-small-v2) and 12.8 percentage points on FAISS (all-MiniLM-L6-v2). The same patterns are also observed at a higher $efSearch$ of 40 (Table 6).

Table 6 shows that despite the overall higher recall, the differences between recall for each insertion order remain comparable. HNSWLib gives a maximum difference of 7.1 percentage points (e5-small-v2) and FAISS gives a maximum difference of 11.5 percentage points (all-MiniLM-L6-v2)

### 5.2.3 LID Ordered Insertion and Relevance.
Recall can be associated with other metrics which influence the efficacy of models on tasks such as information retrieval. Retrieval evaluation for all datasets using PyTREC Eval[41] shows a Pearson correlation coefficient of 0.71 is observed between recall@10 and NDCG@10. Table 7 shows implications in leaderboard ranking of different insertion orders for the HNSWLib implementation.

**Table 5: Average recall@10 with $efSearch = 10$ across benchmark datasets for each model with data inserted in various orders.**

| Model | Desc. LID HNSWLib Recall | Asc. LID HNSWLib Recall | Random Order HNSWLib Recall | Desc. LID FAISS Recall | Asc. LID FAISS Recall | Random Order FAISS Recall |
|---|---|---|---|---|---|---|
| bge-base-en | **0.8498** | 0.8298 | 0.8255 | **0.7760** | 0.6982 | 0.7450 |
| bge-base-en-v1.5 | **0.8664** | 0.8399 | 0.8467 | **0.8046** | 0.7324 | 0.7783 |
| bge-small-en | **0.8461** | 0.8279 | 0.8201 | **0.7819** | 0.7696 | 0.7467 |
| bge-small-en-v1.5 | **0.8771** | 0.8565 | 0.8476 | **0.8007** | 0.7383 | 0.7883 |
| bge-micro | **0.8478** | 0.8199 | 0.8230 | **0.7933** | 0.6991 | 0.7618 |
| stella-base-en-v2 | **0.8801** | 0.8590 | 0.8448 | **0.8288** | 0.7915 | 0.7775 |
| e5-base | **0.8224** | 0.8066 | 0.7603 | **0.7514** | 0.6801 | 0.6854 |
| e5-base-v2 | **0.8171** | 0.7838 | 0.7416 | **0.7461** | 0.6565 | 0.6587 |
| e5-small | **0.8031** | 0.7910 | 0.7611 | **0.7432** | 0.6459 | 0.6819 |
| e5-small-v2 | **0.7943** | 0.7385 | 0.6902 | **0.6963** | 0.6531 | 0.6136 |
| multilingual-e5-base | **0.7720** | 0.7305 | 0.7177 | **0.7000** | 0.6711 | 0.6266 |
| multilingual-e5-large | **0.7982** | 0.7728 | 0.7396 | **0.7165** | 0.6501 | 0.6622 |
| multilingual-e5-small | **0.7762** | 0.7236 | 0.7258 | **0.7033** | 0.6140 | 0.6112 |
| ember-v1 | **0.8754** | 0.8580 | 0.8503 | **0.8325** | 0.7429 | 0.7921 |
| all-MiniLM-L6-v2 | **0.8923** | 0.8650 | 0.8797 | **0.8532** | 0.7250 | 0.8309 |
| gte-base | **0.8884** | 0.8839 | 0.8819 | 0.7747 | 0.8318 | **0.8385** |

**Table 6: Average recall@10 with $efSearch = 40$ across benchmark datasets for each model with data inserted in various orders.**

| Model | Desc. LID HNSWLib Recall | Asc. LID HNSWLib Recall | Random Order HNSWLib Recall | Desc. LID FAISS Recall | Asc. LID FAISS Recall | Random Order FAISS Recall |
|---|---|---|---|---|---|---|
| bge-base-en | **0.9664** | 0.9479 | 0.9585 | **0.9660** | 0.8835 | 0.9529 |
| bge-base-en-v1.5 | **0.9745** | 0.9548 | 0.9662 | **0.9749** | 0.8930 | 0.9630 |
| bge-small-en | **0.9645** | 0.9608 | 0.9588 | **0.9638** | 0.9569 | 0.9546 |
| bge-small-en-v1.5 | **0.9776** | 0.9693 | 0.9688 | **0.9730** | 0.8902 | 0.9678 |
| bge-micro | **0.9653** | 0.9372 | 0.9579 | **0.9659** | 0.8706 | 0.9534 |
| stella-base-en-v2 | **0.9776** | 0.9610 | 0.9683 | **0.9789** | 0.9707 | 0.9654 |
| e5-base | **0.9534** | 0.9442 | 0.9283 | **0.9502** | 0.9140 | 0.9285 |
| e5-base-v2 | **0.9522** | 0.9401 | 0.9197 | **0.9501** | 0.9403 | 0.9140 |
| e5-small | **0.9489** | 0.9293 | 0.9237 | **0.9471** | 0.8330 | 0.9224 |
| e5-small-v2 | **0.9307** | 0.8598 | 0.8535 | **0.9111** | 0.8015 | 0.8664 |
| multilingual-e5-base | **0.9136** | 0.8903 | 0.8945 | **0.9100** | 0.9055 | 0.8906 |
| multilingual-e5-large | **0.9306** | 0.9213 | 0.9149 | **0.9317** | 0.8542 | 0.9137 |
| multilingual-e5-small | **0.9072** | 0.8786 | 0.9038 | **0.9131** | 0.8168 | 0.8533 |
| ember-v1 | **0.9784** | 0.9558 | 0.9626 | **0.9769** | 0.8770 | 0.9692 |
| all-MiniLM-L6-v2 | **0.9829** | 0.9566 | 0.9638 | 0.9790 | 0.8642 | **0.9792** |
| gte-base | **0.9775** | 0.9616 | 0.9140 | **0.9805** | 0.9689 | 0.9718 |

**Table 7: Rank by NDCG at $efSearch = 10$ using HN-SWLib with different insertion orders.**

| Model | Random | Asc. LID | Desc. LID |
|---|---|---|---|
| ember-v1 | 1 | 1 | 1 |
| **gte-base** | 2 | 2 | **3** |
| **bge-base-en-v1.5** | 3 | 3 | **2** |
| stella-base-en-v2 | 4 | 4 | 4 |
| **bge-small-en-v1.5** | 5 | **6** | **6** |
| **bge-base-en** | 6 | **5** | **5** |
| bge-small-en | 7 | 7 | 7 |
| **all-MiniLM-L6-v2** | 8 | 8 | **9** |
| **e5-base-v2** | 9 | 9 | **8** |
| e5-base | 10 | 10 | 10 |
| multilingual-e5-large | 11 | 11 | 11 |
| **bge-micro** | 12 | **15** | **14** |
| **e5-small-v2** | 13 | 13 | **15** |
| **e5-small** | 14 | **12** | **12** |
| **multilingual-e5-base** | 15 | **14** | **13** |
| multilingual-e5-small | 16 | 16 | 16 |

The observations from Table 7 translate into impacts on downstream retrieval tasks. Given that model rankings shift under different insertion orders we can ascertain that each models vector space is not impacted equally; certain models exhibit more robustness to changes in insertion order.

## 5.3 Category Based Insertion Orders and Recall

The insertion sequence of data, influenced by LID, represents a constructed scenario unlikely to mirror the more stochastic nature of real-world data indexing. Nonetheless, practical applications frequently encounter non-random data insertion phenomena. To elucidate the real-world relevance of insertion sequence and its correlation with LID, we examine data from two distinct online retail platforms - one focusing on fashion, the other on homewares.

Contrastive Language-Image Pre-training (CLIP)[9, 19, 35, 39] models are used to generate embeddings from product images, with a series of GPT4[31] generated e-commerce search terms

serving as queries. Specifically the ViT-B-32 architecture with the laion2b_s34b_b79k checkpoint and the ViT-L-14 architecture with the laion2b_s32b_b82k checkpoint were used.

Data was indexed sequentially organized by product categories as listed on the retailers' websites; data within each category is randomly ordered. This approach has similarities with LID-ordered insertion, presupposing that items within the same category exhibit closer proximity to one another than to items from disparate categories. We show that the category ordered insertions studied here have parallels to the LID ordered insertions in section 5.2.2. Analysis of the intrinsic dimensionality, as determined by the PCA method, shows that categories within the data have differing values to each other. Categories exhibit lower intrinsic dimensionalities than the dataset as a whole (Table 8 and Table 9).

**Table 8: Intrinsic Dimensionality in the Fashion Dataset**

| Category | Intrinsic Dim. ViT-B-32 | Intrinsic Dim. ViT-L-14 |
|---|---|---|
| Watches | 312 | 336 |
| Streetwear | 434 | 463 |
| Collectibles | 440 | 463 |
| Sneakers | 389 | 427 |
| Handbags | 418 | 450 |
| **All Data** | **442** | **469** |

**Table 9: Intrinsic Dimensionality in the Homewares Dataset**

| Category | Intrinsic Dim. ViT-B-32 | Intrinsic Dim. ViT-L-14 |
|---|---|---|
| Kitchenware (1) | 394 | 448 |
| Bed (2) | 400 | 453 |
| Pet (3) | 374 | 405 |
| Lighting (4) | 371 | 447 |
| Rugs (5) | 338 | 415 |
| Office (6) | 313 | 369 |
| Lifestyle (7) | 366 | 383 |
| Wall (8) | 421 | 455 |
| Furniture (9) | 383 | 448 |
| Renovation (10) | 396 | 448 |
| Baby (11) | 411 | 442 |
| Home (12) | 426 | 456 |
| **All Data** | **439** | **475** |

In the analysis of the fashion dataset with the search parameter $efSearch = 10$, we observed significant variations in recall based on the order of insertion and the choice of model. Specifically, for ViT-B-32, the recall difference attributable to varied insertion sequences reached up to 7.7 percentage points (Table 8).

**Table 10: Recall at $efSearch = 10$ for Fashion Dataset**

| Order | Model | HNSW & FAISS Avg. Recall |
|---|---|---|
| Hbgs.-Snkrs.-Wtchs.-Coll.-Stwr. | ViT-B-32 | 0.435639 |
| **Snkrs.-Hbgs.-Coll.-Stwr.-Wtchs.** | **ViT-B-32** | **0.512328** |
| Coll.-Stwr.-Hbgs.-Wtchs.-Snkrs. | ViT-L-14 | 0.405639 |
| **Hbgs.-Coll.-Snkrs.-Stwr.-Wtchs.** | **ViT-L-14** | **0.466295** |

This variance in recall metrics shows that the impact of data insertion sequences on the effectiveness of HNSW-based retrieval systems can be observed in real-world applicable scenarios. Moreover, this disparity persists at a higher setting of $efSearch = 40$ (Table 11).

**Table 11: Recall at $efSearch = 40$ for Fashion Dataset**

| Order | Model | HNSW & FAISS Avg. Recall |
|---|---|---|
| Coll.-Hbgs.-Stwr.-Wtchs.-Snkrs. | ViT-B-32 | 0.744918 |
| **Snkrs.-Hbgs.-Coll.-Stwr.-Wtchs.** | **ViT-B-32** | **0.799016** |
| Coll.-Hbgs.-Wtchs.-Stwr.-Snkrs. | ViT-L-14 | 0.712656 |
| **Coll.-Snkrs.-Hbgs.-Stwr.-Wtchs.** | **ViT-L-14** | **0.77741** |

The differences in recall are less significant for the orders attempted with the homewares data, though exhaustively trying every combination of categories was not feasible due to the number of categories. Results for the homewares dataset are shown in Table 12 and Table 13 for $efSearch = 10$ and $efSearch = 40$ respectively - category names are mapped to numbers in Table 9 for brevity.

**Table 12: Recall at $efSearch = 10$ for Homewares Dataset**

| Order | Model | HNSW & FAISS Avg. Recall |
|---|---|---|
| 12-9-1-8-10-2-5-4-11-7-3-6 | ViT-B-32 | 0.668467 |
| **1-2-3-4-5-6-7-8-9-10-11-12** | **ViT-B-32** | **0.69007** |
| 12-9-1-8-10-2-5-4-11-7-3-6 | ViT-L-14 | 0.672265 |
| **1-2-3-4-5-6-7-8-9-10-11-12** | **ViT-L-14** | **0.69547** |

**Table 13: Recall at $efSearch = 40$ for Homewares Dataset**

| Order | Model | HNSW & FAISS Avg. Recall |
|---|---|---|
| 3-11-5-4-1-2-6-7-8-9-10-12 | ViT-B-32 | 0.91251 |
| **1-2-3-4-5-6-7-8-9-10-11-12** | **ViT-B-32** | **0.91979** |
| 12-9-1-8-10-2-5-4-11-7-3-6 | ViT-L-14 | 0.91857 |
| **3-11-5-4-1-2-6-7-8-9-10-12** | **ViT-L-14** | **0.92533** |

# 6 CONCLUSION

In this work we have shown that the construction of HNSW graphs can be sensitive to properties of the datasets and models utilised. The effect of insertion order for data into the graphs has real world impacts, especially in applications where the temporal component of incoming data is correlated with properties of the vector space that the data occupies; such as new product categories being added or domain shifts more generally.

The relationship between intrinsic dimensionality and recall, paired with the relationship between recall and downstream retrieval tasks, indicates that optimal model selection for HNSW based retrieval systems is not as simple as following the results of a benchmark done with exact KNN.

We hope that this work encourages further research into the HNSW algorithm to improve robustness against the insertion order of the data. Other advances may exist in model development as well, allowing for better understanding and control of properties of the vector space which can have impacts on recall in approximate retriever systems.

# 7 FUTURE WORK

It is clear that there exists a relationship between intrinsic dimensionality of vectors, particularly within local neighbourhoods, that has direct impacts on the construction of HNSW graphs. Future work should aim to explore the relationship between intrinsic dimensionality and recall with other approximate retrieval algorithms such as DiskANN[20], FreshDiskANN[40], IVFPQ[24], Random Projection Trees[12] (ANNOY[1]), MRPT[18], and KD-Trees[6] to assess if similar properties are present in these algorithms.

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

Received 25 April 2024
