# OpenReview forum: "The Impacts of Data, Ordering, and Intrinsic Dimensionality on Recall in Hierarchical Navigable Small Worlds"
_ACM.org/SIGIR/ICTIR/2024/Conference — ICTIR 2024_

### Official Review · Reviewer_xh7t · 2024-05-13

**Rating:** 2
**Confidence:** 4

**Objective Part Of Review:**

This is a very interesting experimental work. Many related works have suggested different variants of approximate nearest neighbor search algorithms. However, these works rarely evaluated on contemporary real-world data. Thus, the findings from the experimental evaluations so far might not give a correct picture of the performance of ANN algorithms in practice. This is the core statement of the paper, and this statement is experimentally verified.

I found it especially interesting that the order of data when constructing an ANN-Index matters that much. The authors start from synthetic datasets where they can control the locally intrinsic dimensionality, before they move over to contemporary text retrieval benchmark datasets and CLIP-based search queries on a fashion and homeware dataset.

Overall, the paper is an easy read with a clear result.

**Subjective Part Of Review:**

* All figures and table titles are too short. Especially the titles of the tables should explain what the reader is supposed to see in the data.
* Only HNSW and FAISS are used as ANN algorithms.
* While the paper is an easy read with a clear result, the work is also preliminary. The authors have pointed out, that the LID of the data generated by some modern DNN plays an important role in the ranking of the different ANN algorithm. However, only a few examples are shown. A more systematic approach (e.g, Transformer vs. CNNs vs. RNN, supervised vs. self-supervised learning) is desirable. Also, more about how to order incoming data during index construction should be further explored. Nevertheless, the paper is already inspiring in its current form.

---

### Official Review · Reviewer_iQxQ · 2024-05-15

**Rating:** 0
**Confidence:** 3

**Objective Part Of Review:**

The authors evaluate the impact of vector space dimensionality and vector order insertion with respect to recall when using HNSW ANN.
They leverage a synthetic task using artificial vector spaces in which the intrinsic dimensionality can be controlled, and retrieval tasks using well-established collections.  Moreover, a proprietary collection is also considered.

Through tests with the synthetic data, the authors hypothesize a relation between order insertion and intrinsic dimension, further showing through experimentation that populating the HNSW index following descending local intrinsic dimensionality is superior to random insertion. This was tested on multiple models, multiple collections, and two HNSW codebases (HSNWLib and FAISS).


Comments/concerns:

- Tables 5 and 6: Recall at 10 is too shallow for a first stage retriever. Results should be assessed on deeper cuts (e.g., 100, 1000).

- The technique used to estimate the local intrinsic dimensionality has quadratic complexity. The authors do not state the impact this computation adds to indexing time (e.g., random insertion time vs LID computation + insertion time).

- The authors mention that MS-MARCO vectors are made available (Section 1.1), but Table 2 does not show MS-MARCO being used for evaluation. Why?

**Subjective Part Of Review:**

The problem is relevant, and the experiments provide useful insights. My main concern is whether a Recall@10 is enough to fully assess the impact. The paper could use a re-organization as pages 6 and onwards were hard to follow.

---

### Official Review · Reviewer_1pZU · 2024-05-17

**Rating:** 0
**Confidence:** 2

**Objective Part Of Review:**

The paper provides a survey of popular vector search systems that use the HNSW algorithm, including some of their parameter settings. It primarily studies the performance of the HNSW algorithm on datasets from three different scenarios and explores the relationship between recall rate and LID under different scenarios, as well as the impact of data insertion order on the performance of HNSW. Additionally, the paper also releases a vector dataset covering multiple popular open-source models. Overall, the paper has a clear organizational structure, the scenarios are set reasonably, and the experimental justification is quite thorough. However, there are still some issues that require clarification from the authors.

**Subjective Part Of Review:**

* The paper has creatively studied the performance of the HNSW algorithm under different intrinsic dimensions by constructing synthetic data. However, synthetic data may lack the complexity and diversity of real-world data. It could be questioned whether it is fair to conclude that the recall rate of HNSW decreases with the increase of intrinsic dimensions based solely on synthetic data.

* The paper has conducted studies on some real-world datasets, especially in the scenario of image retrieval. However, there are many other real-world retrieval datasets, such as code retrieval, etc. Exploring only the scenario of product image retrieval may be somewhat limited.

* HNSW is an algorithm that trades off spatial memory for higher efficiency and accuracy. I believe that research on the HNSW algorithm should focus more on how to reduce memory consumption without compromising performance.

---

### Official Review · Reviewer_3m6t · 2024-05-17

**Rating:** 1
**Confidence:** 4

**Objective Part Of Review:**

The article presents a comprehensive experimental analysis of the Hierarchical Navigable Small Worlds (HNSW) algorithm's applications across diverse datasets that are categorized as synthetic, retrieval benchmarks, and real-world datasets. The authors utilize a version of the Average Recall Metric to evaluate the algorithm's performance based on a set of queries and embeddings that are published on Hugging Face. The primary aim of this study is to address the variability of evaluation datasets for the HNSW algorithm and to compare its major implementations—FAISS and HNSWLib—under various parameter settings.
The article is well-written; however, I recommend a thorough revision to address several minor presentation issues. These include formatting inconsistencies such as a white gap before the Introduction, the references being mentioned in a disorderly manner and not in ascending order, and occasional neglect of previously defined abbreviations, such as "approximate nearest neighbors" for example on page 5 (expected to see ANN at this point). Additionally, Table 5 needs restructuring of the titles in the first row, whereas Tables 1, 2, and 3 exhibit misaligned content, diverging from the formatting standard observed in the rest of the tables where elements are aligned in the middle under the column’s title.
A critical correction needed concerns the symbol for Recall@10, which differs from the definition provided in Section 5. In the definition’s formula, a dash above the recall definition seems to represent an average but fails to convey this clearly. Moreover, the rationale behind employing this average recall metric, as opposed to a standard Recall@k and Precision@k, needs clearer justification. The exclusion of an average precision metric Precision(Q,k) similar to the author’s rationale, should be substantiated if it is deemed irrelevant or unhelpful for this analysis. Such an explanation would strengthen the methodological foundation of the study.
Lastly, the discussion lacks an analysis of retrieval times, a critical aspect often considered in retrieval studies. Including this could provide a more comprehensive evaluation of the algorithm's performance across various applications.

**Subjective Part Of Review:**

The article is well-written and demonstrates academic rigor, with mainly minor typographical errors. However, the disordered placement of references complicates the evaluation process. It is recommended that the authors arrange the references in a systematic order.
The research problem posed and addressed is compelling and could be helpful for researchers developing approximate nearest neighbor (ANN) algorithms or the broader community engaged in information retrieval. The article could enhance the community's understanding of parameter tuning in Hierarchical Navigable Small World (HNSW) while highlighting interesting observations regarding ordering and intrinsic dimensionality. Although the methodology employed is not particularly novel, some techniques such as the use of the average recall metric score against an exact retriever and the extensive experiments and discussions provided are likely to be highly valuable for theoretical research.

---

### Meta-Review · Area_Chair_CZY7 · 2024-05-20

**Recommendation:** Reject
**Confidence:** 4

**Metareview:**

The paper presents an insightful examination of the influence of vector space dimensionality and the order of vector insertion on the recall performance of Hierarchical Navigable Small World (HNSW) Approximate Nearest Neighbor (ANN) algorithms. The authors utilize a synthetic task with controllable intrinsic dimensionality alongside retrieval tasks using both well-established and proprietary collections to thoroughly investigate these parameters. However, there are some weaknesses.

1. The evaluation only on the HNSW method, which descends the influence of this paper.
2. The evaluation of this paper is unconvinced. The evaluation results on MS MARCO should be conducted. More evaluation metrics should be conducted to show the performance of the paper.